# DIFFERENTIALLY PRIVATE FINE-TUNING OF LANGUAGE MODELS

**Da Yu**[1,2] **Saurabh Naik**[4] **Arturs Backurs**[3,*] **Sivakanth Gopi**[3,*] **Huseyin A. Inan**[3,*]
**Gautam Kamath**[5,*] **Janardhan Kulkarni**[3,*] **Yin Tat Lee**[3,6,*] **Andre Manoel**[3,*]
**Lukas Wutschitz**[4,*] **Sergey Yekhanin**[3,*] **Huishuai Zhang**[2,*]

[1]Sun Yat-sen University[†], [2]Microsoft Research Asia
[3]Microsoft Research, [4]Microsoft
[5]Cheriton School of Computer Science, University of Waterloo
[6]University of Washington
[1]yuda3@mail2.sysu.edu.cn, [2]huzhang@microsoft.com
[3]{arturs.backurs, sigopi, huseyin.inan}@microsoft.com
[3]{jakul, amonteiroman, yekhanin}@microsoft.com
[4]{snaik, lukas.wutschitz}@microsoft.com
[5]g@csail.mit.edu, [6]yintat@uw.edu

## ABSTRACT

We give simpler, sparser, and faster algorithms for differentially private fine-tuning of large-scale pre-trained language models, which achieve the state-of-the-art privacy versus utility tradeoffs on many standard NLP tasks. We propose a meta-framework for this problem, inspired by the recent success of highly parameter-efficient methods for fine-tuning. Our experiments show that differentially private adaptations of these approaches outperform previous private algorithms in three important dimensions: utility, privacy, and the computational and memory cost of private training. On many commonly studied datasets, the utility of private models approaches that of non-private models. For example, on the MNLI dataset we achieve an accuracy of $87.8\%$ using RoBERTa-Large and $83.5\%$ using RoBERTa-Base with a privacy budget of $\varepsilon = 6.7$. In comparison, absent privacy constraints, RoBERTa-Large achieves an accuracy of $90.2\%$. Our findings are similar for natural language generation when privately fine-tuning GPT-2. Our experiments also show that larger models are better suited for private fine-tuning: while they are well known to achieve superior accuracy non-privately, we find that they also better maintain their accuracy when privacy is introduced.

## 1 INTRODUCTION

Deep learning models are well known to leak sensitive information about the dataset when trained using conventional methods (Shokri et al., 2017; Carlini et al., 2019; 2021). To combat this issue, models can instead be trained to guarantee differential privacy (DP) (Dwork et al., 2006b), a strong notion of data privacy which limits the influence of any individual training point on the final model. While DP is one of the few approaches capable of providing machine learning models with rigorous privacy guarantees, it generally comes at a cost in terms of test accuracy. One oft-cited explanation is that the constraint of DP necessitates much more training data (Tramèr & Boneh, 2021; Feldman, 2020; Brown et al., 2021). Unfortunately, more training data may be hard to acquire, particularly in settings where privacy is a concern.

Parallel to these developments, Transformer-based (Vaswani et al., 2017) large language models (LLMs), including the BERT (Devlin et al., 2019; Liu et al., 2019) and GPT (Radford et al., 2018; 2019; Brown et al., 2020) families, have enabled significant progress in natural language processing, achieving state-of-the-art accuracy in almost every task considered. These models are first pre-trained on an extremely large and diverse public dataset. The weights are then fine-tuned for

---

[*]Authors contribute equally to this work and are listed in alphabetical order.
[†]The work of Da Yu is done while he was an intern at Microsoft Research Asia.

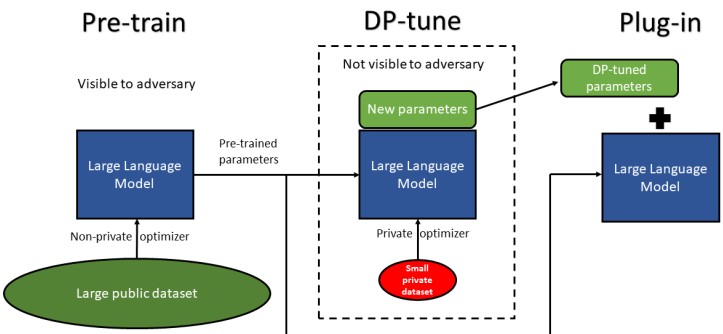

Figure 1: An illustration of our framework. First, the model is pre-trained on a large, public dataset. Next, new parameters are introduced and privately fine-tuned on a smaller, private, task-specific dataset. The original parameters are frozen during this process. Finally, the fine-tuned new parameters may be released publicly and plugged-in to the model for downstream tasks, while still preserving privacy of the private dataset.

Table 1: Accuracy of fine-tuning for downstream tasks with RoBERTa-Large (in %). Our results achieve accuracy comparable to full fine-tuning non-privately, while simultaneously guaranteeing differential privacy. We choose $\delta$ =1e-5 for SST-2 and QNLI and $\delta$ =1e-6 for MNLI and QQP due to their dataset sizes. Implementation details are in Section 4.1.

| Method | MNLI | SST-2 | QQP | QNLI | Avg. | Trained params |
|---|---|---|---|---|---|---|
| Non-private fine-tuning | 90.2 | 96.4 | 92.2 | 94.7 | 93.4 | 100% |
| Our results ($\varepsilon = 6.7$) | 87.8 | 95.3 | 87.4 | 90.8 | 90.3 | 0.94% |

each task of interest using a much smaller task-specific dataset. For example, a single pre-trained GPT-family model may be fine-tuned for various downstream tasks, such as email reply suggestion, sentence completion in text editors, language translation, and more. This two-stage paradigm can naturally be adapted to solve tasks in private learning, automatically addressing the aforementioned data shortage issue via the massive scale of the public pre-training dataset. One may pre-train the model on public data as usual,[1] but then fine-tune the model *privately*.

Despite the success of these models, task-specific fine-tuning introduces a number of technical challenges. In the non-private setting, the immense size of LLMs makes it impractical to fine-tune the full model and store a separate copy of the parameters for hundreds of downstream tasks. Things only get worse with privacy, which leads to overheads in terms of running time, memory usage, and most importantly, accuracy. The magnitude of noise introduced to a model due to DP grows as the model size increases (Bassily et al., 2014; Abadi et al., 2016; Bun et al., 2014), which can overwhelm any signal for larger models. A recent line of work in the non-private literature has proposed parameter-efficient methods to alleviate the issues of storage and computational cost for fine-tuning (Houlsby et al., 2019; Li & Liang, 2021; Aghajanyan et al., 2020; Hu et al., 2021; Mahabadi et al., 2021). The main focus of our work is to explore parameter-efficiency in the context of private learning.

## 1.1 OUR CONTRIBUTIONS

Our primary contribution is to show that advanced parameter-efficient methods can lead to *simpler* and significantly improved algorithms for private fine-tuning. Our framework is illustrated in Figure 1. Our findings and contributions are summarized as follows:

**State-of-the-art utility and privacy.** Empirical evaluation of our algorithms reveals that they achieve state-of-the-art accuracy versus privacy tradeoffs, improving upon the previous best (Yu

---

[1]Despite the fact that the pre-training data is public, there may nonetheless be privacy concerns related to personal or copyrighted data. However, since these pre-trained models have already been released, any associated privacy loss has already been incurred.

et al., 2021b). More importantly, for many fine-tuning tasks, the utility of models trained with DP approaches that of non-private models. For example, privately fine-tuning RoBERTa-Large on the MNLI data set (Williams et al., 2018), we achieve an accuracy of $87.8\%$ with a privacy budget of ($\varepsilon = 6.7, \delta = $ 1e-6). Without privacy guarantees, RoBERTa-Large achieves an accuracy of $90.2\%$ (GPT-3 is known to achieve $91.7\%$ (Hu et al., 2021)); see Table 1 for a summary. We also explore private natural language generation tasks, fine-tuning GPT-2 models on the E2E dataset (Novikova et al., 2017). Again, the utility approaches non-private levels: we achieve a ROUGE-L score of 0.6755 with GPT-2-Large and ($\varepsilon = 5.4, \delta = $ 1e-5), compared to 0.72 without privacy.

**Larger models are better.** Prior work has consistently shown that larger language models achieve better accuracy for downstream tasks. Our results give evidence that this phenomenon extends to the private setting. For example, on the MNLI dataset, RoBERTa-Base achieves an accuracy of $83.5\%$ whereas RoBERTa-Large achieves an accuracy of $87.8\%$, both under a privacy budget of ($\varepsilon = 6.7, \delta = $ 1e-6). Similarly, privately fine-tuning with E2E, GPT-2-Small, GPT-2-Medium, and GPT-2-Large achieve ROUGE-L scores of 0.6219, 0.6645 and 0.6755 respectively, all under a privacy budget of ($\varepsilon = 5.4, \delta = $ 1e-5). While established in the non-private setting, we find this phenomenon quite surprising under DP. There is often a tension when choosing private model architectures: larger models may have higher capacity, but necessitate the introduction of more noise. Consequently, smaller and simpler private models achieve the better accuracy in several settings (Papernot et al., 2019; Tramèr & Boneh, 2021). In contrast, our experiments show that fine-tuning the biggest models achieves the best accuracy.[2]

**Simpler, sparser, and faster.** Beyond accuracy concerns, DP requirements also lead to significant overheads in terms of computation and memory usage. The large number of parameters contributes to the high cost of training LLMs, and things get worse under privacy, which has been documented to increase training time by up to two orders of magnitude (Carlini et al., 2019; Subramani et al., 2021). The parameter-efficient approaches we employ partially offset this issue: as we only update a small fraction of the total number of parameters, training becomes considerably more computationally and memory efficient. Furthermore, as in the non-private setting, this framework leads to a modular design, where a single large pre-trained model can be augmented with lightweight modifications for each individual downstream task.

To the best of our knowledge, we are the first to fine-tune GPT-2-Large using differential privacy, the largest model trained thus far using DP. Given our state-of-the-art results for a variety of standard NLP tasks using advanced fine-tuning techniques, we believe that our paper will serve as a benchmark for further work in this direction. For example, the best average accuracy achieved by the prior work of Yu et al. (2021b) on four standard NLP tasks in Table 1 is $83.9\%$ using $\varepsilon = 8$ (and the same $\delta$ as in Table 1), whereas we can achieve an average accuracy of $90.3\%$ using $\varepsilon = 6.7$ by a combination of better algorithms, larger models, and new privacy accounting techniques.

Finally, though recently considered elsewhere (see Section 5), we put further focus on the framing of public pre-training and private fine-tuning as an important conceptual direction in DP deep learning.

## 2 PRELIMINARIES AND PRIOR ALGORITHM BASELINES

Recall the formal definition of differential privacy.

**Definition 2.1** (Differential Privacy (DP) (Dwork et al., 2006b;a)). *A randomized algorithm* $\mathcal{A}$ *is* ($\varepsilon$,$\delta$)-*differentially private if for any two neighboring datasets* $D$ *and* $D'$, *which differ in exactly the data pertaining to a single user, and for all sets* $\mathcal{S}$ *of possible outputs:* $\Pr[\mathcal{A}(D) \in \mathcal{S}] \leq e^{\varepsilon} \Pr[\mathcal{A}(D') \in \mathcal{S}] + \delta$.

We review prior techniques for private fine-tuning.

---

[2]An alternative perspective is that what we currently think of as "large" language models are relatively small, and we are yet to reach the point where the benefits of model size on accuracy are outweighed by the drawbacks.

## 2.1 Full Fine-tuning via DPSGD

To train a machine learning model with privacy, the most popular algorithm is the celebrated DP stochastic gradient descent (DPSGD) (Song et al., 2013; Bassily et al., 2014; Abadi et al., 2016). This optimization method serves as a drop-in replacement for SGD, augmenting it with the addition of per-example gradient clipping and Gaussian noise addition steps. These two steps serve to limit and mask the contribution of a single example. Two key points to note are that a) per-example gradient clipping incurs significant computational and memory overheads in most implementations, and b) noise introduced due to privacy grows as the square-root as the number of model parameters. With this tool in place, the most basic fine-tuning strategy is to train all parameters using DPSGD.

## 2.2 Reparametrized Gradient Perturbation

To mitigate the limitations of DPSGD, a recent work of Yu et al. (2021b) introduced an elegant method called *reparametrized gradient perturbation* (RGP). RGP exploits the implicit low-rank structure in the gradient updates of SGD to substantially improve upon DPSGD. Specifically, they reparametrize each layer's weight matrix $W$ into $LR + \tilde{W}$, where $L$ and $R$ are low-rank gradient-carrier matrices and $\tilde{W}$ is the residual weight. The authors show that one can obtain a low-dimensional projection of $W$'s gradient by taking gradients only of the low-rank matrices $L$ and $R$ (and not the high-rank $\tilde{W}$). Privacy is introduced by clipping and noising these low-dimensional gradients of $L$ and $R$. While this low-dimensional projection loses some of the signal in $W$'s gradient, it turns out to contain enough to still achieve high accuracy. At the same time, the low-dimensional gradients alleviate the aforementioned issues related to privatization, significantly reducing the memory consumption and noise introduced.

## 3 Our Approach

### 3.1 A Meta-framework

We introduce our approach as a meta-framework for private deep learning, which abstracts the key principles of recent fine-tuning methods.

Suppose $f(W_{\mathrm{PT}}; x)$ is a pre-trained model where $W_{\mathrm{PT}}$ are the pre-trained weights and $x$ is any input. We create a new fine-tuning model

$$f_{\mathrm{FT}}(W_{\mathrm{PT}}, \theta; x) \tag{1}$$

which incorporates additional trainable parameters $\theta$, where $\dim(\theta) \ll \dim(W_{\mathrm{PT}})$. That is, the number of new parameters in $\theta$ is a small fraction of the original number of parameters in the pre-trained weights $W_{\mathrm{PT}}$. Fine-tuning is done by running DPSGD on the additional parameters $\theta$, while freezing the weights of pre-trained model $W_{\mathrm{PT}}$. The new parameters are initialized to $\theta_0$ such that

$$f_{\mathrm{FT}}(W_{\mathrm{PT}}, \theta_0; x) = f(W_{\mathrm{PT}}; x). \tag{2}$$

The initialization condition (2) is very important, as it ensures that fine-tuning starts at the pre-trained model and improves it by modifying the parameters $\theta$. Most fine-tuning methods are additive and have the following special form:

$$f_{\mathrm{FT}}(W_{\mathrm{PT}}, \theta; x) = f(W_{\mathrm{PT}} + \pi(\theta); x), \tag{3}$$

i.e., they modify the pre-trained weights by adding a correction term $\pi(\theta)$ parametrized by $\theta$.

Recent work in the non-private literature has described concrete instantiations of this framework (Houlsby et al., 2019; Mahabadi et al., 2021; Hu et al., 2021), which (crucially) are effective when $\dim(\theta) \ll \dim(W_{\mathrm{PT}})$. In the non-private setting, such reparametrizations are useful for reducing the computation and memory required for fine-tuning, and enable lightweight and plug-in modifications to the base model for different downstream tasks. At the same time, they maintain (or sometimes surpass) the accuracy achieved by full fine-tuning.

We give some intuition as to why parameter-efficient methods can to be more effective for private fine-tuning, especially on smaller datasets. For simplicity, we assume that the fine-tuning method is additive as in (3), such that the fine-tuned weights $W_{\mathrm{FT}} = W_{\mathrm{PT}} + \pi(\theta)$. We can imagine that

$W_{\text{FT}}$ lies on a manifold passing through $W_{\text{PT}}$ of very small dimension (equal to the dimension of $\theta$) compared to the dimension of $W_{\text{PT}}$. Even if the parameters $\theta$ are very noisy due to the noise added during DPSGD, we will always stay in this manifold. In particular, we are not disturbing the pre-trained weights in most directions (those orthogonal to the manifold near $W_{\text{PT}}$). If we run DPSGD on all the weights instead, then we add noise in all directions, thus potentially unlearning the knowledge learned during pre-training, especially in low data regimes. However, this intuition may not always be true; see the remark at the end of our experiments on NLU tasks.

Besides substantial gains in the accuracy, the above method of reparametrization has several other advantages:

- A single pre-trained model such as BERT or GPT is generally applied to hundreds of down-stream tasks via fine-tuning. Private fine-tuning using previous methods requires updating *all* parameters and storing a different copy of the fine-tuned model per task. This creates substantial overheads for storing and deploying, and can be very expensive in practice. On the other hand, the reparametrization (1) means that we only need to store a single pre-trained model that can be shared across many downstream tasks. Each downstream task requires only a small number of new parameters that can be plugged in.

- Differentially private training requires computing and storing per-example gradients, which increases the memory footprint. In our approach, however, learning is done in a much lower dimension, hence saving on the memory cost as compared to prior works.

- Finally, we expect that (1) also gives a more communication-efficient method of fine-tuning in distributed settings such as federated learning, due to the significantly smaller number of parameters learned during fine-tuning.

## 3.2 INSTANTIATING THE META-FRAMEWORK

In this section, we discuss a few ways to instantiate our meta-framework. This list is non-exhaustive, but covers the methods we employ in our experiments.

### 3.2.1 FINE-TUNING VIA LOW-RANK ADAPTATION

Low-Rank Adaptation (LoRA) (Hu et al., 2021) is an additive fine-tuning scheme as defined in (3). For each dense weight matrix $W_{\text{PT}}^i$ of size $a \times b$ in the pre-trained network, we add a low-rank correction term $L^i R^i$, i.e.,

$$W^i = W_{\text{PT}}^i + L^i R^i, \tag{4}$$

where $L^i \in \mathbb{R}^{a \times r}, R^i \in \mathbb{R}^{r \times b}$ are new trainable parameters. Hu et al. (2021) apply this reparameterization only to the Transformer attention weights ($W_q, W_v$), and freeze all other weights (e.g., $W_k$ and $W_o$ and those in the feed-forward layers). The rank $r$ is typically chosen to be small, e.g., $r = 4, 16, 64$. Since most parameters in Transformer architectures are dense weight matrices, choosing a small $r$ results in a nearly square-root reduction in the number of parameters.

### 3.2.2 FINE-TUNING VIA ADAPTERS

Houlsby et al. (2019) propose adapter-based fine-tuning, in which we modify the architecture of the pre-trained model by adding new "adapter" layers after each attention and feed-forward layer. Adapter layers are bottleneck layers with residual connections. Specifically, given an input $x$, an adapter layer $A$ performs

$$A(x) = U(\tau(D(x))) + x, \tag{5}$$

where $U$ is an up-projection affine linear map, $D$ is a down-projection affine linear map, and $\tau$ is a non-linear activation function such as the Gaussian error Linear Unit (GeLU) (Hendrycks & Gimpel, 2016). If $x$ has dimension $d$, then $U \in \mathbb{R}^{d \times r}, D \in \mathbb{R}^{r \times d}$ for some $r \ll d$. Thus, the number of introduced parameters is significantly less than the number of parameters in the pre-trained model. When fine-tuning, the parameters of the original model are frozen, and only parameters of the adapter layers, as well as layer normalizations, are modified. Note that fine-tuning with adapters is not an additive fine-tuning framework as in (3), but is captured by the broader framework in (1).

Table 2: Memory and speed comparison for RoBERTa-Large. The rank is chosen as $r = 16$ for RGP and LoRA. The speed is measured by the wall-clock time for training one epoch of the SST-2 dataset on a single Tesla V100 GPU with gradient accumulation for batch size 2000.

| Method | Memory (GB) | Speed (seconds per epoch) |
|---|---|---|
| Full fine-tuning (DPSGD) | 27.9 | 715 |
| RGP | 9.1 | 296 |
| DP LoRA | 6.1 | 271 |

### 3.2.3 FINE-TUNING VIA COMPACTER

The recent work of Mahabadi et al. (2021) introduces Compacters (Compact adapters), a method which further improves the parameter efficiency of adapters. This is done by replacing the dense matrices in the up-projection $U$ and down-projection $D$ by tensor products of smaller matrices, thus reducing the number of trainable parameters. Specifically, they replace the dense matrix $M_\ell$ in the adapter layer $\ell$ by a low-rank parameterized hypercomplex multiplication (LPHM) layer, i.e., each dense matrix $M_\ell \in \mathbb{R}^{a \times b}$ is expressed as

$$M_\ell = \sum_{i=1}^{n} A_i \otimes \left( S_i^\ell T_i^\ell \right) \tag{6}$$

where $A_i \in \mathbb{R}^{n \times n}, S_i^\ell \in \mathbb{R}^{a/n \times k}, T_i^\ell \in \mathbb{R}^{k \times b/n}$ and $\otimes$ is the matrix Kronecker product. Note the matrices $A_i$ are not indexed by the layer $\ell$ because these matrices are shared among all the adapter layers. Since each adapter layers has two dense matrices (one for up-projection and one for down-projection), if there are $L$ adapter layers, this reduces the number of parameters from $L(2ab)$ to $L(2(a + b)k) + n^3$. In practice, $a$ and $b$ are chosen to be either the model dimension $d$ or the intermediate representation dimension $r$ in the adapters, $n$ is typically chosen to be a small constant such as $n = 2, 4, 8, 12$ and $k$ is chosen to be $1$.

### 3.2.4 WHY DOES PARAMETER-EFFICIENT TUNING WORK?

Theoretical explanation of success of parameter-efficient fine-tuning methods is active area of research in deep learning. Indeed, since trends have consistently shown that model accuracy increases with size, how can one achieve competitive accuracy while fine-tuning less than $1\%$ of the parameters? One popular hypothesis is *intrinsic dimensionality* (Li et al., 2018), which posits that the minimum number of parameters needed to train a machine learning model may be much less than the total number of model parameters. Aghajanyan et al. (2020) explore this hypothesis in the context of fine-tuning LLMs, showing that one can achieve most of their accuracy by training only a very small number of parameters (chosen via a random projection). Perhaps surprisingly, they find that as *the model size increases, intrinsic dimension decreases*, in the limit exhibiting zero-shot learning. While we did not explore this hypothesis in the context of DP due to computational restrictions, we believe it may be an interesting lens through which one can understand the effectiveness of private parameter-efficient fine-tuning.

### 3.3 COMPARISION WITH BASELINE ALGORITHMS

We highlight some key algorithmic differences between our proposed methods and the baselines of full fine-tuning and RGP.

- DPSGD and RGP both require updating all parameters of the pre-trained model, whereas our proposed methods update only a tiny fraction (between $0.05\%$ and $1\%$). The rightmost columns of Tables 3 and 4 list the number of parameters trained by these algorithms.

- RGP performs a low-rank decomposition of weight matrices which is similar to LoRA, though there are subtle differences. Recall that in RGP, at the beginning of each iteration $t$, the historical weight matrix $W_{t-1}$ is decomposed to find a low-rank product $LR$. The gradients on $L$ and $R$ are then projected back to the full parameter space to perform the descent step. Hence, RGP keeps modifying the pre-trained weights during learning.

  LoRA can be viewed as a simplification of RGP. LoRA reparametrizes $W_{\mathrm{FT}} := W_{\mathrm{PT}} + LR$, where the pre-trained weight matrix $W_{\mathrm{PT}}$ is frozen during training. Hence, compared to

Table 3: Accuracy for fine-tuning with RoBERTa-Base (in %). The privacy parameters are $\varepsilon = 6.7$, and $\delta =$ 1e-5 for SST-2 and QNLI and 1e-6 for MNLI and QQP. Bold indicates the best accuracy with DP. Numbers for non-private fine-tuning are from Liu et al. (2019) and Hu et al. (2021).

| Method | | MNLI | SST-2 | QQP | QNLI | Avg. | Trained params |
|---|---|---|---|---|---|---|---|
| Full | w/o DP | 87.6 | 94.8 | 91.9 | 92.8 | 91.8 | 100% |
| | DP | 53.1 | 82.6 | 74.4 | 63.9 | 68.5 | |
| LoRA | w/o DP | 87.5 | 95.1 | 90.8 | 93.3 | 91.7 | 0.24% |
| RGP[3] | DP | 80.1 | 91.6 | 85.5 | 87.2 | 86.1 | 100% |
| Adapter | DP | 83.4 | **92.5** | 85.6 | **87.5** | **87.3** | 1.4% ($r = 48$) |
| Compacter | DP | 82.6 | 92.3 | 84.7 | 85.1 | 86.2 | 0.055% ($r = 96, n = 8$) |
| LoRA | DP | **83.5** | 92.2 | **85.7** | 87.3 | 87.2 | 0.94% ($r = 16$) |

RGP, LoRA eliminates the decomposition and the projection to the full parameter space at each iteration, simplifying the implementation and reducing the running time and memory cost. This is summarized in Table 2. We observe that DP LoRA reduces the memory cost by about 33% and the training speed by 8%. As we will see, this simplification also results in improved utility.

- Neither full fine-tuning nor RGP fall into our meta-framework described by (1). Thus, if a pre-trained model is to be applied to several downstream tasks, one must store a separate set of weights for each task, incurring a significant memory cost and losing the plug-in functionality. In contrast, our methods are much more lightweight.

## 4 EXPERIMENTS

We experimentally evaluate our methods for DP fine-tuning to demonstrate their utility, privacy, and parameter-efficiency. We investigate both language understanding and text generation tasks to establish that our techniques are applicable to a variety of tasks and model architectures. Our code is publicly available at https://github.com/AnonymousAKES/Differentially-Private-Fine-tuning-of-Language-Models.

### 4.1 FINE-TUNING FOR LANGUAGE UNDERSTANDING TASKS

We first compare our methods with state-of-the-art fine-tuning algorithms using models from the BERT family, which was used in the prior work (Yu et al., 2021b). Specifically, we use RoBERTa models (Liu et al., 2019), which are pre-trained on public data collected from the web. RoBERTa-Base has 125M parameters and RoBERTa-Large has 355M parameters. We choose four downstream tasks: MNLI, QQP, QNLI, and SST-2 from GLUE (Wang et al., 2018), following Yu et al. (2021b).

**Implementation Details:** For fine-tuning with adapters, we may choose the intermediate representation dimension $r$, shared across all adapter layers. For fine-tuning with Compacter, we can choose both $r$ and the Kronecker product kernel dimension $n$ in (6). For LoRA fine-tuning, we add bottleneck branches for both the attention layers and the feedforward layers, which differs slightly from the addition of bottleneck branches for only the $W_q$ and $W_v$ matrices of the attention layers as done by Hu et al. (2021). Given the same bottleneck representation dimension $r$ in (4), our new implementation uses twice as many trainable parameters as the original paper, and achieves some improvements for learning with DP. We perform privacy accounting using the approach of Gopi et al. (2021), which currently gives the tightest bounds.

**Hyperparameter choice:** Given the large number of hyperparameter choices, e.g., the intermediate representation dimension, learning rate, weight decay, privacy parameter $\delta$, and model size, an exhaustive grid search over all hyperparameters is expensive. Our hyperparameter choices are informed by prior work and are as follows. For privacy parameters, we use $\delta = $ 1e-5 for SST-2 and QNLI and $\delta = $ 1e-6 for QQP and MNLI due to their dataset sizes, and use noise multipliers $0.92, 0.83, 0.66$ and $0.65$ for SST-2, QNLI, QQP, and MNLI, respectively, which is the same as Yu et al. (2021b). In Appendix B, we run experiments under different privacy parameters. The proposed framework performs well under a wide range choices of $\varepsilon$ and $\delta$. The clipping threshold is 10 for

---

[3]Numbers are from https://github.com/dayu11/Differentially-Private-Deep-Learning.

Table 4: Accuracy for fine-tuning with RoBERTa-Large (in %). The privacy parameters are $\varepsilon = 6.7$, and $\delta =$ 1e-5 for SST-2 and QNLI and $\delta =$ 1e-6 for MNLI and QQP. Bold indicates the best accuracy with DP. Numbers for non-private fine-tuning are from Liu et al. (2019) and Hu et al. (2021).

| Method | | MNLI | SST-2 | QQP | QNLI | Avg. | Trained params |
|---|---|---|---|---|---|---|---|
| Full | w/o DP | 90.2 | 96.4 | 92.2 | 94.7 | 93.4 | 100% |
| LoRA | w/o DP | 90.6 | 96.2 | 91.6 | 94.9 | 93.3 | 0.23% |
| RGP | DP | 86.1 | 93.0 | 86.7 | 90.0 | 88.9 | 100% |
| Adapter | DP | 87.7 | 93.9 | 86.3 | 90.7 | 89.7 | 1.4% ($r = 48$) |
| Compacter | DP | 87.5 | 94.2 | 86.2 | 90.2 | 89.5 | 0.053% ($r = 96, n = 8$) |
| LoRA | DP | **87.8** | **95.3** | **87.4** | **90.8** | **90.3** | 0.94% ($r = 16$) |

all methods. The batch size is 2000. In Appendix D, we show the performance of the proposed algorithm is stable across a wide range of choices of clipping thresholds and batch sizes. For adapters and Compacter, we follow the original papers and choose $r$ from $\{16, 48, 96\}$ and $n$ from $\{4, 8, 12\}$. For LoRA, we choose the best-performing rank $r$ from the set $\{4, 16, 48, 64\}$. The best performing hyperparameters are noted in Tables 3 and 4. We train for 20 epochs using AdamW (Loshchilov & Hutter, 2019) with weight decay 1e-2 and search over four learning rates $\{$5e-4, 1e-3, 2e-3, 5e-3$\}$.

**Results:** We report the prediction accuracy in Tables 3 and 4. Our experiments using RoBERTa-Base serve as a direct comparison to Yu et al. (2021b) who only trained the base model, whereas RoBERTa-Large experiments demonstrate the significance of using larger models. Our key findings are: **(1)** On *all* datasets, our methods achieve the best accuracy while training a tiny fraction of parameters; larger models give significant improvements. **(2)** Noticeable improvements in $\varepsilon$ versus Yu et al. (2021b) are primarily due to new privacy accountants based on Fourier-based numerical composition (Koskela et al., 2020; 2021; Gopi et al., 2021); we use the accountant in Gopi et al. (2021) since it is the most efficient. **(3)** Private adapters provide the best average performance for RoBERTa-Base, whereas LoRA outperforms all other methods for RoBERTa-Large.

**Remark:** While our experiments indicate that full fine-tuning does not achieve competitive performance, there could be a choice of hyperparameters that improves upon the reported numbers, e.g., "mega" batch sizes (in the millions) in Anil et al. (2021). We note that our main message is that one does not need to fine-tune all parameters to achieve the best accuracy. Nevertheless, it is interesting to wonder if full fine-tuning with DPSGD can match the accuracy of parameter-efficient methods. A positive answer would imply that private and non-private fine-tuning conceptually mirror each other.

**Update:** A concurrent work by Li et al (Li et al., 2022) show that using a larger batch size and training with full-precision improves the performance of full fine-tuning via DPSGD, and obtains similar performance as our algorithms. Thus, poor performance of DPSGD in our experiments is due to the suboptimal choice of hyperparameters and also due to precision issues, although we use same hyperparameters for all the algorithms. We run new experiments with hyperparameters of (Li et al., 2022) in full precision mode, and get improvements around 1% even for our algorithms. We report these findings in Appendix C.

## 4.2 Fine-tuning for Natural Language Generation (NLG)

Next, we study private fine-tuning for text generation problems using the GPT-2 series of models on the End-2-End (E2E) NLG challenge (Novikova et al., 2017), one of the primary benchmarks used in recent works on non-private fine-tuning (Hu et al., 2021; Li & Liang, 2021). We use GPT-2-Small (117M parameters), GPT-2-Medium (345M parameters), and GPT-2-Large (774M parameters).[4] To the best of our knowledge, we are the first to privately fine-tune for E2E or fine-tune GPT-2-Large. The purpose of this section is not to evaluate various fine-tuning algorithms, but to show that private fine-tuning is competitive with non-private fine-tuning for text generation problems. Due to the high cost of training, we report experimental results only for fine-tuning with LoRA.

In Appendix F, we present additional experiments on NLG that include private fine-tuning of the GPT-2-XL model with **1.5 billion** parameters. Other noticeable points in the additional experiments

---

[4]https://huggingface.co/transformers/model_doc/gpt2.html.

Table 5: Metrics on the E2E NLG task ($\varepsilon = 5.4, \delta =$1e-5). Non-DP results from Hu et al. (2021).

| Method | BLEU | NIST | MET | ROUGE-L | CIDEr |
|---|---|---|---|---|---|
| GPT-2-Small + DP | 59.26 | 6.13 | 36.6 | 64.1 | 1.63 |
| GPT-2-Medium + DP | 64.2 | 7.77 | 40.02 | 66.45 | 2.00 |
| GPT-2-Large + DP | 64.51 | 8.22 | 41.5 | 67.55 | 2.13 |
| GPT-2-Medium | 70.4 | 8.85 | 46.8 | 71.8 | 2.53 |
| GPT-2-Large | 70.4 | 8.89 | 46.8 | 72.0 | 2.47 |

include 1) we show improved performance using better hyperparameters; 2) we test different privacy parameters; 3) we consider a new dataset DART (Nan et al., 2021).

**E2E NLG challenge:** The E2E dataset in Novikova et al. (2017) contains template-like information in the restaurant domain to be mapped to natural language with end-to-end training. The dataset consists of 42K training samples, 4.6K validation samples, and 4.6K test samples. We use standard metrics such as BLUE, ROUGE-L, etc., used in (Hu et al., 2021) for evaluation.

**Hyperparameter choice:** For LoRA, we choose the bottleneck rank $r = 4$ in (4) and fine-tune $W_q$ and $W_v$ matrices of the attention layers as in the original paper. We optimize using AdamW with learning rate 2e-4, weight decay 1e-2 and train our models for 5 epochs using batch size 64. We take the gradient clipping parameter to be 1.0 and set the noise multiplier as 0.5.

**Results:** The results of our experiments are summarized in the Table 5, which reiterate the main themes of this paper: private fine-tuning with a parameter-efficient approach performs close to their non-private counterparts and show consistent improvement in the utility as the model size increases.

## 5 RELATED WORK

Some work studies private language models on more traditional architectures such as LSTMs (Hochreiter & Schmidhuber, 1997), either training with DPSGD (McMahan et al., 2018; Carlini et al., 2019) or related heuristics (Ramaswamy et al., 2020). Though pre-training on public data is suggested (McMahan et al., 2018), public data appears to only be used in one of these works for honest hyperparameter selection (Ramaswamy et al., 2020). A few more recent works consider training LLMs with DP. Anil et al. (2021) privately train BERT-Large from scratch, compared to our work which focuses on private fine-tuning. (Hoory et al., 2021; Basu et al., 2021) perform private full fine-tuning of BERT models. Hoory et al. (2021) achieve accuracy which is comparable to the non-private model, but additionally supplement the public pre-training data with additional domain-relevant material, while we use off-the-shelf pre-trained models. Basu et al. (2021) observe significant drops in utility, compared to our parameter-efficient methods which do not. While Kerrigan et al. (2020) consider public pre-training and private fine-tuning, their experiments are on much smaller architectures (i.e., feedforward networks with three hidden layers). A simultaneous work of Ginart et al. (2022) investigates private *prediction* (rather than learning) for next-token prediction. A subsequent work by Senge et al. (2021) also investigates the effect of private fine-tuning on various NLP tasks.

In a concurrent work, Li et al. (2022) also investigate DP fine-tuning of LLMs. In several cases, their results demonstrate qualitatively similar findings as ours. While our experiments focus primarily on parameter-efficient fine-tuning methods, interestingly, they show that private full fine-tuning can also achieve comparable utility if the experimental setup is configured properly, e.g., using suitable hyperparameters. In Appendix C, we run experiments under the setup in Li et al. (2022). We show their setup can also improve the performance of our methods.

## 6 CONCLUSION

So far, DP deep learning has focused on training models from scratch. The spectacular success of transfer learning in real-world applications, however, shows that private fine-tuning is an equally pertinent problem to study and deserves more attention. We show that by combining recent advances in NLP, parameter-efficiency, privacy accounting, and using larger models, one can privately fine-tune models whose utility approaches that of non-private models. We hope our work inspires more study on the core problem of private fine-tuning, which we believe to be a central direction for research in private machine learning, leading to more interaction between the LLM and DP communities.

## ACKNOWLEDGMENTS

The authors would like to thank Rabeeh Karimi Mahabadi for sharing hyperparameters based on experiments in Mahabadi et al. (2021). Janardhan Kulkarni would like to thank Edward Hu for sharing many ideas on fine-tuning. Gautam Kamath is supported by an NSERC Discovery Grant.

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

Table 6: Test accuracy for fine-tuning RoBERTa-Large with different privacy parameters. The number of training samples is denoted by $n$. The values of $\sigma$ are noise multipliers. Numbers in the brackets are the changes compared to the results in Table 4 ($\varepsilon = 6.7$, $\delta = \Theta(1/n)$).

| Taks | $\sigma$ | $\delta = 1/n$ | $\delta = 1/10n$ | $\delta = 1/100n$ | $\delta = 1/1000n$ | Accuracy (in %) |
|---|---|---|---|---|---|---|
| MNLI | 1.88 | $\varepsilon = 1$ | $\varepsilon = 1.35$ | $\varepsilon = 1.49$ | $\varepsilon = 1.61$ | 86.8 (-1.0%) |
| QQP | 1.88 | $\varepsilon = 1$ | $\varepsilon = 1.40$ | $\varepsilon = 1.54$ | $\varepsilon = 1.67$ | 85.2 (-2.2%) |
| QNLI | 3.01 | $\varepsilon = 1$ | $\varepsilon = 1.48$ | $\varepsilon = 1.64$ | $\varepsilon = 1.79$ | 88.0 (-2.8%) |
| SST-2 | 3.63 | $\varepsilon = 1$ | $\varepsilon = 1.47$ | $\varepsilon = 1.64$ | $\varepsilon = 1.80$ | 93.1 (-2.2%) |
| MNLI | 0.91 | $\varepsilon = 3$ | $\varepsilon = 4.12$ | $\varepsilon = 4.51$ | $\varepsilon = 4.89$ | 87.4 (-0.4%) |
| QQP | 0.93 | $\varepsilon = 3$ | $\varepsilon = 4.10$ | $\varepsilon = 4.49$ | $\varepsilon = 4.86$ | 86.8 (-0.6%) |
| QNLI | 1.29 | $\varepsilon = 3$ | $\varepsilon = 4.45$ | $\varepsilon = 4.90$ | $\varepsilon = 5.33$ | 89.9 (-0.9%) |
| SST-2 | 1.52 | $\varepsilon = 3$ | $\varepsilon = 4.37$ | $\varepsilon = 4.83$ | $\varepsilon = 5.25$ | 94.1 (-1.2%) |

## A  ADDITIONAL RELATED WORK

There exist other parameter-efficient tuning methods which we did not evaluate in our work. Some of these include random subspace projection (exploiting intrinsic dimensionality (Li et al., 2018; Aghajanyan et al., 2020)), prefix and prompt tuning (Li & Liang, 2021; Lester et al., 2021), tuning only biases (Cai et al., 2020; Ben Zaken et al., 2021), and other architecture variants including Adapters (Pfeiffer et al., 2021; Rücklé et al., 2020). An interesting direction for future work is to see whether parameter-efficient tuning approaches specifically designed for the private setting can achieve higher utility. We also mention zero-shot learning, in which no task-specific dataset is required and thus perfect privacy is achieved. Currently, zero-shot approaches achieve low utility compared to fine-tuning, though it is possible that future models may narrow this gap.

Finally, our investigation fits more broadly into a line of work employing public data for private data analysis. Some works on image classification consider pre-training on a large public dataset and fine-tuning on a smaller private dataset (Abadi et al., 2016; Papernot et al., 2019; Tramèr & Boneh, 2021; Luo et al., 2021). In particular, Luo et al. (2021) investigate the role of parameter efficiency in private fine-tuning ResNet models, and propose strategies to choose which parameters to fine-tune. One line of work uses unlabeled public data to train a student model (Papernot et al., 2017; 2018; Bassily et al., 2018), including one work simultaneous to our own for natural language generation Tian et al. (2022). Another recent idea uses a small amount of public data to identify a lower-dimensional subspace of the gradients in which to perform private descent (Zhou et al., 2021; Yu et al., 2021a; Kairouz et al., 2021). A simultaneous work of Amid et al. (2021) uses public data in the mirror map for a private mirror descent algorithm. Finally, other works (both theoretical and experimental) investigate the role of public data in private query release, synthetic data generation, and prediction (Ji & Elkan, 2013; Beimel et al., 2016; Alon et al., 2019; Nandi & Bassily, 2020; Bassily et al., 2020a;b; Liu et al., 2021).

## B  EXPERIMENTS WITH DIFFERENT PRIVACY PARAMETERS

Now we test our framework under different privacy constraints. Specifically, we run LoRA on the language understanding tasks with various choices of privacy parameters $\varepsilon$ and $\delta$. We consider both RoBERTa-Base and RoBERTa-Large.

For the RoBERTa-Large model, we set $\varepsilon = 1$ and 3 with $\delta$ being the same as those in Section 4. We use the PRV accountant (Gopi et al., 2021). After getting the noise multipliers, we also reduce the value of $\delta$ and report the corresponding value of $\varepsilon$. The hyperparameters are the same as those in Section 4. We run experiments on all four tasks, i.e., MNLI ($n \sim 392k$), QQP ($n \sim 364k$), QNLI ($n \sim 104k$), and SST-2 ($n \sim 67k$). We report the results in Table 6. The performance of our framework is decent even with very tight privacy budgets. For instance, with $\varepsilon < 2$ and $\delta = 1/1000n$, the accuracy gap between the non-private baseline is only 3.8 for MNLI and 2.1 for SST-2.

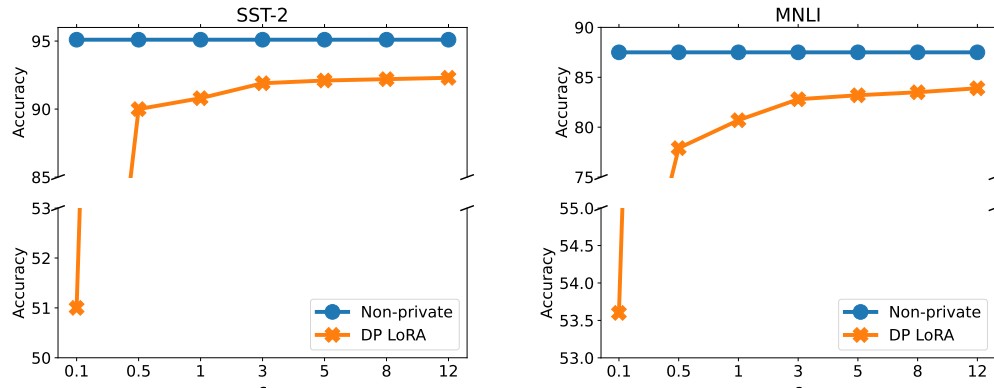

Figure 2: Test accuracy (in %) of fine-tuning the RoBERTa-Base model on MNLI and SST-2 with various choices of $\varepsilon$.

Table 7: Accuracy for fine-tuning downstream tasks with RoBERTa-Base (in %). Experiments are run with full-precision. We also scale up the batch size according to the dataset size compared to SST-2. The privacy parameters are $\varepsilon = 6.7$, and $\delta$ =1e-5 for SST-2 and QNLI and 1e-6 for MNLI and QQP.

| Method | | MNLI | SST-2 | QQP | QNLI | Average Accuracy |
|---|---|---|---|---|---|---|
| Full | w/o DP | 87.6 | 94.8 | 91.9 | 92.8 | 91.8 |
| | DP | 83.2 | 85.9 | 86.2 | 84.8 | 85.0 |
| Adapter | DP | **84.6** | **92.9** | **87.4** | **89.2** | **88.5** |
| LoRA | DP | 84.5 | 92.7 | 87.1 | 88.3 | 88.2 |

For the RoBERTa-Base model, we try various choices of $\varepsilon$. The values of $\varepsilon$ are chosen from $[0.1, 0.5, 1, 3, 5, 8, 12]$. All other settings are the same as those in Section 4. We run experiments on the MNLI and SST-2 datasets. The results are presented in Figure 2. Our framework performs well for a wide range of $\varepsilon$. We note that our algorithm achieves meaningful accuracy even for very tight privacy parameters $\varepsilon = 0.5$ and 1. Such values of $\varepsilon$ are rarely explored when training deep models with differential privacy.

## C  FINE-TUNING FOR LANGUAGE UNDERSTANDING TASKS WITH LARGE BATCH SIZE AND FULL-PRECISION

Li et al. (2022) show the performance of fine-tuning the full model can be significantly improved with proper configuration. In this section, we re-evaluate the tasks in Table 3 and 4 under the configuration in Li et al. (2022) and show such a configuration also improves the performance of our methods.

The configuration in Li et al. (2022) has two major differences compared to that in Section 4.1. The first difference is Li et al. (2022) run experiments with full-precision while the experiments

Table 8: Accuracy for fine-tuning downstream tasks with RoBERTa-Large (in %). Experiments are run with full-precision. We also scale up the batch size according to the dataset size compared to SST-2. The privacy parameters are $\varepsilon = 6.7$, and $\delta$ =1e-5 for SST-2 and QNLI and $\delta$ =1e-6 for MNLI and QQP.

| Method | | MNLI | SST-2 | QQP | QNLI | Average Accuracy |
|---|---|---|---|---|---|---|
| Full | w/o DP | 90.2 | 96.4 | 92.2 | 94.7 | 93.4 |
| | DP | 86.4 | 90.9 | 87.5 | 89.4 | 88.6 |
| Adapter | DP | 88.6 | 94.5 | 87.8 | 91.6 | 90.6 |
| LoRA | DP | **89.0** | **95.3** | **88.4** | **92.4** | **91.3** |

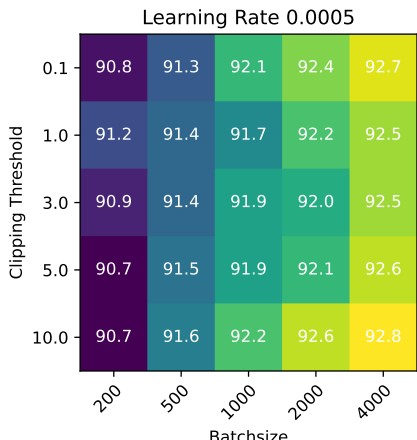 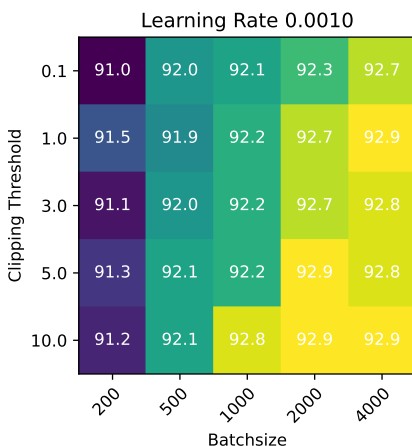

Figure 3: Test accuracy (in %) of fine-tuning RoBERTa-Base with differentially private LoRA on the SST-2 dataset. Our algorithm performs well on a wide range of hyperparameters.

Table 9: Non-private metrics on the E2E NLG task, using full fine-tuning.

| Method | BLEU | NIST | MET | ROUGE-L | CIDEr |
|---|---|---|---|---|---|
| GPT-2-Medium | 68.2 | 8.62 | 46.2 | 71.0 | 2.47 |
| GPT-2-Large | 68.5 | 8.78 | 46.0 | 69.9 | 2.45 |

in Section 4.1 use half-precision. Using half-precision is a common approach to speed up NLP experiments (Ott et al., 2018). However, half-precision may incur underflow issue which impacts the model performance (Micikevicius et al., 2017). The second difference is they use larger batch size for larger datasets. For example, the batch size for MNLI is roughly six times larger than the batch size for SST-2 in Li et al. (2022). In Section 4.1, we use the same batch size for all datasets.

We follow the above setup and re-evaluate DP-LoRA and DP-Adapter. The results are in Table 7 and 8. The results of full fine-tuning with differential privacy are directly adopted from Li et al. (2022). The configuration in Li et al. (2022) further improves the strong results in Table 3 and 4. For example, we achieve 89.0% accuracy on the MNLI dataset, which is only 1.2% lower than the accuracy without DP constraint. Moreover, the benefit of the proposed framework over full fine-tuning is still clear. The average accuracy of the proposed algorithms is ∼3% higher than that of full fine-tuning.

## D    ON THE INFLUENCE OF HYPERPARAMETERS

Here we demonstrate that our algorithms perform well for a wide range of hyperparameters. We study two hyperparameters that are directly related to the variance of noise: clipping threshold and batchsize. The clipping threshold is chosen from $[0.1, 1.0, 3.0, 5.0, 10.0]$ and the batchsize is chosen from $[200, 500, 1000, 2000, 4000]$. We note that we keep the number of updates the same as that in Section 4 when the batchsize is changed. We fine-tune the RoBERTa-Base model with differentially private LoRA ($r = 16$) on the SST-2 dataset. The results are presented in Figure 3. DP LoRA performs well for all the hyperparameters considered. The gap between the best accuracy and the worst accuracy is only 2%.

## E    FULL FINE-TUNING WITH GPT-2

All results in Table 5 (in the main body), both private and non-private, perform fine-tuning using LoRA. In Table 9, we additionally report utility of non-private full fine-tuning. These numbers are taken from Table 1 of Li & Liang (2021). In general, these numbers are slightly lower than those obtained by performing non-private fine-tuning with LoRA.

Table 10: Metrics on the E2E NLG task. Non-DP results from Hu et al. (2021), except for GPT-2-XL, which was not reported in the paper. We ran GPT-2-XL with hyperparameters presented in Hu et al. (2021). Bold indicates the best accuracy with DP. DP parameters are ($\varepsilon = 6.0, \delta = $ 1e-5). Val perp stands for validation perplexity.

| Method | Val perp | BLEU | NIST | MET | ROUGE-L | CIDEr |
|---|---|---|---|---|---|---|
| GPT-2-Small + DP | 4.51 | 63.8 | 7.19 | 39.5 | 67.5 | 1.87 |
| GPT-2-Medium + DP | 4.02 | 65.5 | 8.45 | 42.7 | 67.9 | 2.23 |
| GPT-2-Large + DP | 3.87 | **66.7** | **8.63** | **44.0** | 67.8 | **2.33** |
| GPT-2-XL + DP | **3.79** | 66.1 | 8.53 | 43.0 | **68.1** | 2.28 |
| GPT-2-Medium | 3.19 | 70.4 | 8.85 | 46.8 | 71.8 | 2.53 |
| GPT-2-Large | 3.06 | 70.4 | 8.89 | 46.8 | 72.0 | 2.47 |
| GPT-2-XL | 3.01 | 69.4 | 8.78 | 46.2 | 71.5 | 2.49 |

Table 11: Metrics on the E2E NLG task. Bold indicates the best accuracy with DP. DP parameters satisfy ($\varepsilon = 3.0, \delta = $ 1e-5), ($\varepsilon = 3.4, \delta = 1/10n$), ($\varepsilon = 3.9, \delta = 1/100n$) and ($\varepsilon = 4.5, \delta = 1/1000n$). Val perp stands for validation perplexity.

| Method | Val perp | BLEU | NIST | MET | ROUGE-L | CIDEr |
|---|---|---|---|---|---|---|
| GPT-2-Small + DP | 4.59 | 62.7 | 7.03 | 39.2 | 66.4 | 1.85 |
| GPT-2-Medium + DP | 4.08 | 65.2 | 8.31 | 42.2 | 68.1 | 2.22 |
| GPT-2-Large + DP | 3.92 | 66.7 | 8.60 | 43.6 | 68.1 | 2.29 |
| GPT-2-XL + DP | **3.85** | **67.6** | **8.64** | **44.9** | **68.6** | **2.36** |

# F    ADDITIONAL EXPERIMENTS ON NATURAL LANGUAGE GENERATION

In this section, we perform additional experiments on private fine-tuning for text generation problems using the GPT-2 series of models. This includes the private fine-tuning of the GPT-2-XL model with 1.5B parameters. There are three main points to note compared to our results in the main body: 1) We show an improved performance on E2E NLG challenge using better hyperparameters; 2) We conduct experiments on E2E dataset with different privacy parameters to show that large language models perform strong even with smaller privacy budgets; 3) Finally, we conduct new experiments on DART dataset.

## F.1    IMPROVING THE PERFORMANCE ON E2E NLG CHALLENGE

We improve the results of Table 5 with the following set of new hyperparameters.

**Hyperparameter choice:** For LoRA, we choose the bottleneck rank $r = 4$ in (4) and fine-tune $W_q$ and $W_v$ matrices of the attention layers as in the original paper. We optimize using AdamW with learning rate 4e-4, weight decay 1e-2 and train our models for 20 epochs. We use batch size 128. We take the gradient clipping parameter to be 1.0 and the noise multiplier to be 0.6 for the accountant in Gopi et al. (2021), achieving $\varepsilon = 6.0, \delta = $1e-5.

**Results:** The results of our experiments are summarized in the Table 10.

## F.2    EXPERIMENTS WITH DIFFERENT PRIVACY PARAMETERS

On E2E dataset, we test our framework with smaller privacy budgets ($\varepsilon < 5$ and $\delta \ll 1/n$) where $n$ is the number of samples in the training data.

**Hyperparameter choice:** The hyperparameter choices are similar as in Section F.1. The only difference is that we increase the noise multiplier to be 0.71 for the accountant in Gopi et al. (2021), achieving the following $(\varepsilon, \delta)$ pairs: ($\varepsilon = 3.0, \delta = $1e-5), ($\varepsilon = 3.4, \delta = 1/10n$), ($\varepsilon = 3.9, \delta = 1/100n$) and ($\varepsilon = 4.5, \delta = 1/1000n$).

**Results:** The results of our experiments are summarized in the Table 11.

There are a couple of interesting observations comparing Table 11 with Table 10. First, we observe that although privacy budget is tight in Table 11, the results are quite similar to Table 10, which

Table 12: Metrics on the DART dataset. Non-DP results from Hu et al. (2021), except for GPT-2-XL, which was not reported in the paper. We ran GPT-2-XL with hyperparameters presented in Hu et al. (2021). Bold indicates the best accuracy with DP. DP parameters are ($\varepsilon = 6.8, \delta = $ 1e-5). Val perp stands for validation perplexity. Unlike all other metrics, the lower the TER metric is the better for the performance of the model.

| Method | Val perp | BLEU | MET | TER |
|---|---|---|---|---|
| GPT-2-Small + DP | 3.82 | 38.5 | 0.34 | 0.53 |
| GPT-2-Medium + DP | 3.30 | 42.0 | 0.36 | 0.51 |
| GPT-2-Large + DP | 3.10 | 43.1 | 0.36 | **0.5** |
| GPT-2-XL + DP | **3.00** | **43.8** | **0.37** | **0.5** |
| GPT-2-Medium | 2.67 | 47.1 | 0.39 | 0.46 |
| GPT-2-Large | 2.89 | 47.5 | 0.39 | 0.45 |
| GPT-2-XL | 2.83 | 48.1 | 0.39 | 0.46 |

shows that our methods also perform very well under stronger privacy guarantees. A more interesting observation is that under smaller epsilon regimes, the performance for private fine-tuning of GPT-2-XL model improves. Observe that the performance improvement is more prominent going from GPT-2-Small to GPT-2-XL in this setting, which may indicate that larger models can be even more effective in private learning when the privacy budgets are tight.

### F.3 PERFORMING EXPERIMENTS ON DART DATASET

We study the DART dataset as a text generation problem for private fine-tuning of GPT-2 series of models.

**DART:** DART was introduced as an open-domain data-to-text dataset by Nan et al. (2021). The dataset consists of 62K training samples, 6.9K validation samples, and 12K test samples. In comparison to E2E, the dataset is larger and the task is more challenging.

**Hyperparameter choice:** The hyperparameter choices are similar as in the previous setting. The only difference is that we use batch size 256 for the experiments on DART. This achieves $\varepsilon = 6.8, \delta = $1e-5 on DART using the accountant in Gopi et al. (2021).

**Results:** The results of our experiments are summarized in the Table 12.

