# OpenReview forum: "Differentially Private Fine-tuning of Language Models"
_ICLR.cc/2022/Conference — ICLR 2022 Poster_

### Official Review · Reviewer_oFqs · 2021-10-31

**Correctness:** 3
**Technical Novelty And Significance:** 3
**Empirical Novelty And Significance:** 3
**Recommendation:** 6
**Confidence:** 4

**Main Review:**

Strengths
1. The paper is well written and organized. The framework and the particular methods were explained clearly and the experiments were well executed, proving the claims in the paper.
2. The proposed framework could have a lot of impact for NLP practitioners when training DP. Large models are notoriously difficult to train with DP from scratch. Given there are huge amounts of public texts and pretrained models in the NLP community, fine-tuning with privacy as proposed in this paper could be standard for learning on sensitive data.

Weaknesses and Questions
1. Technique novelty is a bit limited given that all the methods (Adapters, Compactors and LoRA) in the proposed framework are from prior non-private fine-tuning works. The only thing different is that this paper is more using these methods with DPSGD are effective. Could there be specific changes that are more effective just for private learning?
2. Adapters, Compactors and LoRA are also not only parameter-efficient methods. How well might other sparsification techniques (e.g. lottery tickets [1], diff pruning [2]) compare to the proposed framework?
3. Baseline comparison is lacking, although the authors provided the non-private utility baseline, all non-private numbers are learned with full SGD. It would make more sense to provide non-private numbers for the corresponding parameter-efficient methods.
4. The comparison for RGP is also a bit unfair because it was originally proposed for training with privacy from scratch. Is it likely that training large models from scratch with DP could be better with the parameter-efficient frameworks as well?

References

[1] The Lottery Ticket Hypothesis: Finding Sparse, Trainable Neural Networks. Jonathan Frankle, Michael Carbin

[2] Parameter-Efficient Transfer Learning with Diff Pruning. Demi Guo, Alexander M. Rush, Yoon Kim

**Summary Of The Paper:**

This paper introduces a parameter-efficient framework for privately fine-tuning large language models. The techniques were proposed in prior literature for non-private learning while the authors found that these techniques are very effective for training with differential privacy (DP), which was extremely hard for large models due to the amount of noise added. On common NLP benchmarks and with relatively small privacy budget, these parameter-efficient methods outperformed the regular DPSGD and could even be close to the non-private baseline, in terms of utility. These methods are also much more memory-efficient and faster than DPSGD as well.


**Summary Of The Review:**

Overall, I am leaning towards accepting this paper given the strong results and the hope for learning large models with differential privacy. The experiment section could be strengthened with better comparison to baseline methods.

---

> ### Author Response · Authors · 2021-11-15
> **Response to Reviewer oFqs**
>
> We thank the reviewer for their careful reading and thoughtful comments. We address all their concerns below. Some experiments and changes have been added to the appendix to make them easy to find, which would be incorporated as appropriate into the body of the final version of the paper. In summary, the experiments in the appendix consider smaller epsilon values, larger models (GPT-2-XL with 1.5 billion parameters), DART dataset for generation tasks, and the effect of hyperparameters.
>
> 1. Please see our separate comment regarding the novelty and significance of our work.
>
> 2. There are indeed a number of other excellent parameter-efficient methods for fine-tuning, and we believe that they would display qualitatively similar behavior to the approaches we tested. We highlighted some in the related work, we will additionally include those that the reviewer has mentioned. We note that lottery tickets are not immediately applicable in a similar fashion, as the lottery ticket approach (implemented naively) requires one to first train the network non-privately.
>
> 3. We note that our existing GPT-2 experiments (Table 5) compare non-private fine-tuning with LoRA to private fine-tuning with LoRA. We have further included results for non-private full fine-tuning of GPT-2 in Table 7 in the appendix. We have augmented Tables 3 and 4 to include the accuracy of non-private fine-tuning with LoRA. Similar non-private results for non-private fine-tuning with adapters and Compacter are qualitatively similar and already appear in the literature; we are happy to add further baselines to the final paper as requested.
>
> 4. We agree that RGP can be used for privately training from scratch (as well as some other methods we explore, e.g., LoRA), as done by the authors of the RGP paper (Yu et al., 2021) for vision tasks. However, in the context of LLMs, Yu et al. (2021) employ RGP exclusively for private fine-tuning, matching the setting we consider in our paper. Thus, we take this as evidence of a fair comparison with their method.
>
> Regarding training from scratch, the reviewer raises an interesting question. However, we are unclear on what training from scratch with some of the parameter-efficient methods would look like. For example, adapters could be placed in a Transformer-based architecture, but without the pre-training, the remaining weights would be frozen at initialization (which may be just random values). We are unaware of any exploration of this approach even in the non-private setting, but we conjecture that utility would be low. We further note that DP pre-training of LLMs (as compared to fine-tuning, which we consider) seems to require qualitatively different methods (https://arxiv.org/abs/2108.01624).

---

> > ### Author Response · Authors · 2021-11-25
> > **Baselines**
> >
> > Hi reviewer oFqs, just wanted to point out that we have added more baselines as requested. Are these what you were hoping to see, or are there other baselines that you think would help improve the paper? Thanks!

---

### Official Review · Reviewer_eqgr · 2021-11-02

**Correctness:** 3
**Technical Novelty And Significance:** 2
**Empirical Novelty And Significance:** 2
**Recommendation:** 6
**Confidence:** 5

**Main Review:**

Strength:
+ The problem the paper aims to address is an important one. Machine learning with differential privacy is a very active research area, and this paper is a valuable addition to the literature in this field and could serve as a benchmark for further research in the area.

+ The paper is well-written. The presentation is clear. It is a great pleasure to read through the manuscript.

+ The experiments are well designed and executed. The results show improved privacy/utility trade-off over previous work.

Weakness:
- One major question I have is novelty. The idea to fine-tune large LMs on private datasets is not new [1]. So is the idea to hold most of the pretrained weights constant and only update a small subset of the parameters (e.g., [3]). Thus, this paper is a combination of existing state-of-the-art techniques on several fronts including LoRA [3] for dimension reduction and FFT based method (4) for privacy accounting. While this is solid engineering work and demonstrates the feasibility of integrating DP into NLP, I am not sure if it has enough novelty to be published as a separate research paper.


References:
[1] Differentially Private Language Models Benefit from Public Pre-training

[2] Parameter-efficient transfer learning for nlp.

[3] LoRA: Low-Rank Adaptation of Large Language Models

[4] Numerical composition of differential privacy


**Summary Of The Paper:**

This paper demonstrates the feasibility of fine-tuning large language models (pretrained on public dataset) on private datasets for various downstream tasks. It proposes a meta-framework in which most of the pretrained weights are held constant and only a small number of additional parameters are updated during fine-tuning. Combined with state-of-the-art dimension reduction and privacy accounting techniques, their method achieves good privacy/utility tradeoff on several GLEU benchmarks using RoBERTa and NLG using GPT-2.

**Summary Of The Review:**

This paper presents solid engineering work for adopting DP in NLP tasks using large transformer-based language models pretrained on public datasets. Although the result is interesting, the paper is a combination of several existing techniques and do not have enough novelty to be published as a separate research paper.

---

> ### Author Response · Authors · 2021-11-15
> **Response to Reviewer eqgr**
>
> We thank the reviewer for their careful reading and thoughtful comments. Before we address specific comments, we would like to bring to your notice that some experiments and changes have been added to the appendix to make them easy to find, which would be incorporated as appropriate into the body of the final version of the paper. In summary, the experiments in the appendix consider smaller epsilon values, larger models (GPT-2-XL with 1.5 billion parameters),  DART dataset for generation tasks, and the effect of hyperparameters. Our findings further strengthen the three contributions mentioned in the paper. The two key (new) points to note are: 1) Under smaller values of epsilon, larger models perform even better, 2) Using the SST-2 dataset, we demonstrate that our algorithms achieve comparable accuracy using a broad range of hyperparameter choices (Appendix C). The gap between best and worst accuracy is at most 2%. These new findings give hope for private hyperparameter transfer learning, simultaneously solving the problems of privacy leakage and the computational cost of hyperparameter search.
>
> We note that, to our understanding, [1] does not actually fine-tune LLMs privately, they only privately fine-tune small models and non-privately fine-tune GPT-2. To highlight our discussion in the related work section: "While Kerrigan et al. (2020) consider public pre-training and private fine-tuning, their experiments are on much smaller architectures (i.e., feedforward networks with three hidden layers)." Please see our separate comment regarding the novelty and significance of our work.
>
> We observe that you have chosen the following evaluation for Correctness of our work: "1: The main claims of the paper are incorrect or not at all supported by theory or empirical results." We are unclear which aspects of your review reflect these concerns. If you could clarify, we would be happy to address them.

---

### Official Review · Reviewer_BZJU · 2021-11-02

**Correctness:** 4
**Technical Novelty And Significance:** 2
**Empirical Novelty And Significance:** 3
**Recommendation:** 8
**Confidence:** 4

**Main Review:**

Paper studies the problem of DP finetuning large language models, where DP is enforced during finetuning by using various methods (LoRA, RGP, Compacter). paper is very clearly written and easy to follow.

Although one can say that the paper is mostly empirical and focused on showing that DP finetuning works when the curse of dimensionality for DPSGD is dealt with in one way or the other, I think it adds good value and a proof to the literature.

I do not have any major concerns with the paper.

**Summary Of The Paper:**

Paper studies the deferentially private fine-tuning of large language models and shows that privately finetuning language models can provide good utility.

**Summary Of The Review:**

I like the paper and think that it will add good value to the literature and the conference.

---

> ### Author Response · Authors · 2021-11-15
> **Response to Reviewer BZJU**
>
> We thank the reviewer for their reading and encouraging comments. Some new experiments have been added to the appendix to make them easy to find, which would be incorporated as appropriate into the body of the final version of the paper. In summary, the experiments in the appendix consider smaller epsilon values, larger models (GPT-2-XL with 1.5 billion parameters),  DART dataset for generation tasks, and the effect of hyperparameters, which further support our three main contributions.

---

### Official Review · Reviewer_hxyL · 2021-11-03

**Correctness:** 4
**Technical Novelty And Significance:** 2
**Empirical Novelty And Significance:** 4
**Recommendation:** 6
**Confidence:** 4

**Main Review:**

Pros,
1. The proposed method can handle various fine-tuning algorithms.
2. The experimental results seem promising since the gaps between the proposed methods and the non-protection methods are not so large.
3. The authors conduct experiments on multiple NLP tasks, including NLU and NLP tasks.
4. The framework is also adaptive to other NLG methods.

Cons,
1. Table 3 & 4 compare the proposed methods with the traditional usage (Full). "Full" acts as a baseline method. However, in Table 5 (NLG tasks), there's no baseline method for comparison.
2. It will be better if the model can achieve good performnace when \espilon is less than 5. Usually, \espilon in a range of 0.1 ~ 5 can provide meaningful private protection. Also, the \delta is large than it should be. The value of \delta on the order of 1/|D_train| are very dangerous according to [2]. In that level of \delta, they permit “preserving privacy” by publishing the complete records of a small number of database participants. (But in the experiments of E2E, the data size is 42000 and the \delta is 1/50000.) Unsuitable \epsilon and \delta can ensure the utility (model performance) but the privacy protection becomes too weak.
3. The authors claim that "we are the first to fine-tune GPT-2-Large using differential privacy". Actually, [2] also fine-tune on large GPT-2 with DP-SGD.
4. There's some hyper-parameters play an important role for model training. But, tuning hyper-parameters requires additional information about the private information (accessing validation set or testing set), which leads to private leakage. So, how to count the information leakags of users? How to avoid it?
5. This paper propose a meta-framework that handles several existing fine-tuning algorithms with DP-SGD. The performance is amazing. However, the contribution in terms of the technical novelty is not much.

[1] The Algorithmic Foundations of Differential Privacy, 2014.

[2] Differentially Private Language Models Benefit from Public Pre-training. 2020.

**Summary Of The Paper:**

In this paper, the authors propose a meta-framework that applies DP-SGD to NLP tasks. As a DP learning algorithm, DP-SGD provides a practical method to protect the privacy of the training samples of the deep learning model. The framework works on the fine-tuning phase of the pre-trained language model and protects the privacy of the fine-tuning datasets. To adapt DP-SGD to NLP models, the authors propose to fine-tune only a part of the parameters in the large pre-trained language model. The proposed meta-framework can handle various fine-tuning algorithms. The experimental results show the proposed method can achieve performance very close to the non-protection algorithms.

**Summary Of The Review:**

This paper proposes a meta-framework carrying some existing fine-tune methods. The performance looks good but there are still some issues. The technical novelty is limited.

The rebuttal solved most of my concerns. Thanks for the author's efforts!

---

> ### Author Response · Authors · 2021-11-15
> **Response to Reviewer hxyL**
>
> We thank the reviewer for their careful reading and thoughtful comments. We address all their concerns below. Some experiments and changes have been added to the appendix to make them easy to find, which would be incorporated as appropriate into the body of the final version of the paper. In summary, the experiments in the appendix consider smaller epsilon values, larger models (GPT-2-XL with 1.5 billion parameters), DART dataset for generation tasks, and the effect of hyperparameters.
>
> 1. Full fine-tuning of GPT-2 for E2E NLG has been previously considered (Li and Liang, 2021). We report their numbers in Appendix D, Table 7. Non-private full fine-tuning provides comparable (but slightly less) utility versus non-private fine-tuning with LoRA.
>
> 2. The reason behind the choice of (epsilon, delta) in our submission is to facilitate direct comparison with the previous best results (Yu et al., 2021) on GLUE benchmarks. They reported an epsilon = 8 whereas we get epsilon = 6.8 for the same noise parameters as we use more advanced privacy accounting. The values of delta are the same in both works. To demonstrate the effect of stronger privacy protection with smaller values of epsilon and delta, we report new experimental results in Appendix B, Table 6, and Figure 2. We show that the average drop in utility on GLUE benchmarks is around 2% even with very small epsilon values (i.e., epsilon < 2). We observe similar behavior with GPT-2 on NLG tasks; see Appendix E.3 Table 10. We further note that the performance improvement with larger models is even more pronounced in the high privacy (i.e., small epsilon) regime, further emphasizing our finding that larger models are better suited for private fine-tuning.
>
> 3. As far as we could tell [2] only fine-tune GPT-2 (of an undisclosed size) non-privately. To quote their paper: "Also, we finetune OpenAI’s pre-trained GPT-2 (Radford et al., 2019b) non-privately on both Brown and Reddit." It further appears that they only performed private fine-tuning for very small models (described in their Appendix A). Please let us know if we are mistaken.
>
> 4. The potential for privacy leakage via hyperparameter tuning is a very good point, and we will make it clear in the final version of the paper that we do not account for this loss. We emphasize that almost every paper in the private ML literature takes the perspective of focusing on accuracy under the best choice of hyperparameters. For a fair comparison, we take a similar perspective.
>
>
> Nevertheless, we agree that understanding the cost of hyperparameter selection in DPSGD is an important open problem in private learning. In this direction, we empirically demonstrate on the SST-2 dataset that our algorithms achieve comparable accuracy using a broad range of hyperparameter choices (Appendix C). The gap between best and worst accuracy is at most 2%.
>
> Our new algorithms give hope for private hyperparameter transfer learning, simultaneously solving the problems of privacy leakage and the computational cost of hyperparameter search. We plan to update our paper highlighting these observations to further draw attention of the community towards solving the problem you mentioned.
>
> 5. Please see our separate comment regarding the novelty and significance of our work.

---

### Author Response · Authors · 2021-11-15
**Comment to all readers**

In this general comment, we highlight two separate points. The first is regarding significance and technical novelty of our paper. The second highlights some of our new experiments conducted since the original submission, which also address some specific reviewer concerns. These experiments appear in the appendix of the updated submission.

Technical Novelty of our paper:  It is true that our paper builds upon ideas recently developed in NLP and deep learning literature. On the other hand, we are the first to employ many of these ideas in the context of differentially private machine learning. With this perspective, there are several innovations:

1) Our paper is the first to explore parameter efficiency in the context of DP deep learning, and how it circumvents well-known computational overheads of private training. Parameter efficient techniques will play a significant role in training models at the scale of GPT-3 or GPT-2-XL.

2) We achieve the state of the art numbers for several GLUE benchmarks, improving average accuracy by ~5% while simultaneously decreasing epsilon by ~1.5 compared to prior work published at ICML 2021 (https://arxiv.org/abs/2106.09352).

3) Our findings bring to light a rather surprising phenomenon in the context of DP learning. We show that training larger models achieves a better utility versus privacy tradeoff. Our new experiments show that this phenomenon is even more pronounced in the high privacy regime as shown by our experiment in Appendix E.3 (Compare Table 10 with Table 8). These findings go against the conventional wisdom in the DP community, which suggests that large models have worse utility due to the magnitude of noise introduced scaling with the number of parameters. To us, this is arguably the most important contribution of our work, with the potential of spurring more research in both theory and practice. We would love to know your thoughts on this point.

We believe that these are quite significant conceptual contributions worthy of a broader audience.

Beyond these problem-specific innovations, our work showcases that private deep learning is comparable to non-private deep learning. We were able to train GPT-2-XL (1.5 billion parameters) with minimal loss in accuracy and overhead in training cost. Given the right platform, our work will bring this area of research to the forefront of the NLP community, engaging a broader audience, and hence infusing both the communities with new opportunities. This will ultimately lead to better private systems benefiting everyone, as privacy in deep learning is an important and relevant problem with a broad social impact.

New Experiments: We would also like to highlight additional experiments done since submission to further investigate our algorithms, which also addresses some reviewer comments. The changes have been added to the appendix to make them easy to find, which would be incorporated as appropriate into the body of the final version of the paper. In summary, the experiments in the appendix consider smaller epsilon values, even larger models (GPT-2-XL with 1.5 billion parameters), DART dataset for NLG, and the effect of hyperparameters. Our findings further strengthen the three contributions mentioned in the paper. Some key points to note are: 1) The loss in accuracy due to values of smaller values of epsilon (in the range 2-5) is minimal (around is between 1-2%) for GLUE benchmarks. Same holds for E2E; some of the scores even improve due to better hyper parameters. 2) For smaller values of epsilon (i.e., stronger privacy guarantees), larger models perform even better compared to smaller models. 3)  Using the SST-2 dataset, we show that our algorithms achieve comparable accuracy using a broad range of hyperparameter choices. The gap between the best and worst accuracy is at most 2% for GLUE benchmarks. These new findings give hope for private hyperparameter transfer learning, simultaneously solving the problems of privacy leakage and computational cost in hyperparameter search.

---

### Decision · Program_Chairs · 2022-01-20

**Decision:**

Accept (Poster)

**Comment:**

Discussions and additional baseline experiments added during the author response period were enough to motivate multiple reviewers to change their recommendation to an accept during the author response. Multiple reviewers felt that the technical novelty of the work was limited, but the rebuttal cleared up their concerns enough to motivate them to switch their assessments to accept.

The claim of this work is that it provides a simpler, sparser, and faster algorithms for differentially private fine tuning of LLMs, yielding SOTA privacy results vs. utility on a number of standard NLP tasks. The work proposes a meta-framework.

In the end, all reviewers rated this paper as an accept and the AC also recommends acceptance.